# Positive Mining in Graph Contrastive Learning

## Abstract

Graph Contrastive Learning (GCL), which aims to capture representations from unlabeled graphs, has made significant progress in recent years. In GCL, InfoNCE-based loss functions play a crucial role by ensuring that positive node pairs—those that are similar—are drawn closer together in the representational space, while negative pairs, which are dissimilar, are pushed apart. The primary focus of recent research has been on refining the contrastive loss function, particularly by adjusting the weighting of negative nodes. This is achieved by changing the weight between negative node pairs, or by using node similarity to select the positive node associated with the anchor node. Despite the substantial success of these GCL techniques, there remains a belief that the nodes identified as positive or negative may not accurately reflect the true positives and negatives. To tackle this challenge, we introduce an innovative method known as Positive Mining Graph Contrastive Learning (PMGCL). This method consists in calculating the probability of positive samples between the anchor node and other nodes using a mixture model, thereby identifying nodes that have a higher likelihood of being true positives in relation to the anchor node. We have conducted a comprehensive evaluation of PMGCL on a range of real-world graph datasets. The experimental findings indicate that PMGCL significantly outperforms traditional GCL methods. Our method not only achieves state-of-the-art results in unsupervised learning benchmarks but also exceeds the performance of supervised learning benchmarks in certain scenarios.

## 1 Introduction

In recent years, Graph Neural Networks (GNNs) (Kipf & Welling, 2016; Manessi et al., 2020) have emerged as a powerful class of models for learning representations from graph-structured data. GNNs often demonstrate remarkable performance across various domains by aggregating neighborhood information multiple times, including node classification, link prediction, and graph classification tasks. Traditional GNNs are primarily built upon supervised or semi-supervised approaches, inherently relying on large amounts of high-quality labeled data. However, in practical applications, acquiring an abundance of graph labels requires considerable resources and time (Dai et al., 2022; Shi et al., 2024; Xia et al., 2022; 2021b; Zheng et al., 2022). Consequently, unsupervised learning remains a challenging endeavor.

Contrastive learning (CL) has emerged as a powerful paradigm for unsupervised representation learning. Unlike traditional supervised learning methods that rely on labeled data, contrastive learning leverages the inherent structure and relationships within the data to learn meaningful representations without the need for explicit labels. It has gained a lot of attention and achieved impressive results in various fields, including Computer Vision (CV) (Zhu et al., 2020), Natural Language Processing (NLP) (Aberdam et al., 2021), and more recently Graph Contrastive Learning (GCL) (Hassani & Khasahmadi, 2020; You et al., 2021; Zhu et al., 2020; You et al., 2020), which combines CL with GNN to learn rich information from unlabeled graph data.

A similar process is adopted in existing GCL methods. First, different graph augmentation methods are used to generate various views, such as node dropping (You et al., 2020), edge perturbation (Veličković et al., 2018), attribute masking (Zhu et al., 2021), subgraph sampling (Yang et al., 2022) and graph noise injection (Hassani & Khasahmadi, 2020). Then use the same GNN encoder or a

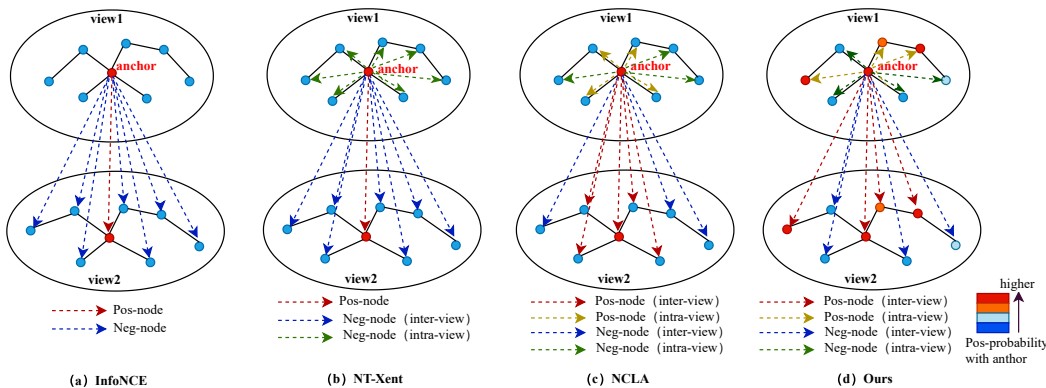

Figure 1: In different node-node contrastive loss positive and negative pairs are selected differently.In (a) and (b), it is shown that InfoNCE and NT-Xent have only one positive pair and multiple negative pairs, while (c) represents the loss function of NCLA, and (d) our proposed PMGCL, both have multiple positive pairs. The red nodes in view1 represent anchors and the entire black line represents the original edge in the network. The dashed lines of different colored arrows represent the positive and negative pairs corresponding to the anchors, and the nodes of different colors represent the probabilities of the positive samples corresponding to the anchors.

different GNN encoder (Yang et al., 2022) to learn the embedding representation for different augmentation views, and finally apply various contrastive loss functions such as InfoNCE (Oord et al., 2018; Zhu et al., 2020), normalized temperature-scaled cross-entropy loss (NT-Xent) (Zhu et al., 2020), extract core information between different augmentation view embedded representations according to InfoMax (Linsker, 1988) principles. Although GCL has made significant achievements, it still has some shortcomings in the selection of contrastive objectives.

Most of the existing GCL methods directly apply the contrastive loss function proposed in CV to graph data (You et al., 2020; Zhu et al., 2020), ignoring the intrinsic differences between images and graphs. In negative mining techniques such as InfoNCE and NT-Xent loss functions. As shown in Figure 1(a) and 1(b), by creating different augmentation views, each anchor forms a positive pair, InfoNCE treats all other distinct nodes of different views as negative pairs, while NT-Xent treats all distinct nodes in the same and different views as negative pairs. Based on this, many GCL methods produce different loss functions by adjusting the weights between negative pairs. This causes nodes belonging to the same classes to be pushed away from the anchor. However, according to Contrastive Learning theory and empirical analysis, samples of the same class should be close to each other, not pushed apart (Tian et al., 2020). As shown in Figure 1(c), NCLA (Shen et al., 2023) solves this problem to some extent. According to the homogeneity hypothesis (McPherson et al., 2001), interconnected nodes usually belong to the same class. Therefore, NCLA takes the neighbor nodes of the anchor as positive samples. In fact, there are many false positives in the neighbor nodes, which will push the positives away inappropriately. Choosing more and the right negatives remains a challenge. To remedy the aforementioned limitations, we propose a new GCL method, called PMGCL, which we believe can distinguish between true and false positives by fitting a two-component (true-false positives) beta mixing model (BMM) (Gupta & Nadarajah, 2004; Ji et al., 2005). With BMM, we can obtain more suitable positives according to the probability of true positives of the anchor, and the loss function is different from the contrast loss function originally proposed by CV (such as InfoNCE and NT-Xent), which only takes one positive pair. We allow multiple positive pairs obtained from BMM. As shown in Figure 1(d), node color represents the probability of being directly associated with the anchor, with red representing the highest probability and blue the lowest probability. The difference from Figure 1(c) is that in our method, neighbor nodes can be negative and non-neighbor nodes can be positive. Our contributions can be summarized as follows:

1) We propose that using BMM to estimate the probability of other nodes being true positive to the anchor is a more suitable method for selecting positive pairs.

2) Instead of applying the contrastive loss in CV to the graph data, we use the new contrastive loss that allows multiple positive pairs per anchor.

3) Our approach provides a significant improvement over GCL's approach. On the node classification task, PMGCL consistently outperforms the state-of-the-art results on multiple unsupervised datasets and even surpass the performance of supervised benchmarks, and we have also achieve promising results on the node clustering task.

## 2 RELATED WORK

### 2.1 GRAPH CONTRASTIVE LEARNING (GCL)

Contrastive learning, as an effective unsupervised learning paradigm, can get rid of the constraints of artificial labels (Hendrycks et al., 2019; Tan et al., 2021) (Xia et al., 2021a). Initially, DGI (Veličković et al., 2018) applies the idea of Deep InfoMax (DIM) to the graph, encoding the local neighborhood of each node and encoding the global graph to learn the representation of nodes. Inspired by DGI, InfoGraph (Sun et al., 2019) uses information sharing between local features of nodes and the global structure of graphs to improve node representation learning. Similarly, GMI (Peng et al., 2020) works by maximizing mutual information between input and output graphs. MV-GRL (Hassani & Khasahmadi, 2020) proposes to learn node-level and graph-level representations by node diffusion and comparing nodes with representations of augmentation graphs. Later, GraphCL (You et al., 2020) proposed different combinations of graph augmentations, including random node drop, feature masking, edge perturbation, subgraph sampling and graph noise injection. To make GraphCL more flexible, JOAO (You et al., 2021) automatically selects combinations of different random graphs for augmentation. Recently, GCL has focused on fully parametric graph augmentation, with AutoGCL (Yin et al., 2022) building a learnable graph generator that learns a probability distribution to help adaptive drop nodes and mask features. SimGRACE (Xia et al., 2021a) even simplifies GCL by removing data augmentations. In this paper, we consider how to select positives to further improve the effectiveness of positive selection in node-level contrastive learning.

### 2.2 CONTRASTIVE OBJECTIVE

Common contrast modes in GCL are graph-graph, graph-node, node-node (Liu et al., 2022). In the node-node GCL method, the positives are close to each other and the negatives are far away from each other. For example, DGI (Veličković et al., 2018) and GMI (Peng et al., 2020) contrasts the neighborhood characteristics and hidden representations of each node. Recently, proposed ProGCL (Xia et al., 2021b), the weight of negative samples is reassigned by mining hard negatives. However, we believe that these methods are all similar to InfoNCE and NT-Xent, taking only one positive and the rest of the nodes are negative, and then pushing it away from the anchor. However, this is not desirable in terms of graph domain, as it may push nodes of the same label away as well. In a recent study, NCLA (Shen et al., 2023) and gCool (Li et al., 2022) considered different approaches to defining positives. NCLA considers all of the anchor's neighbors as positives, and gCool considers all of the nodes in the same community as positives which is not quite appropriate in the real graph data. There are still a large number of negative nodes in the neighborhood or community nodes, resulting in mistakenly pulling the negative nodes closer to the anchor. In this paper, we calculate the probability that nodes and anchors are positives to obtain more reasonable positives and increase the number of positive pairs.

## 3 METHOD

### 3.1 PRELIMINARY

Let $G = (V, E)$ to be a graph, with its node set $V = \{\nu_1, \nu_2, \ldots \nu_n\}$ and edge set $E \subseteq V \times V$. Additionally, $X \in R^{N \times F}$ and $A \in \{0, 1\}^{N \times N}$ are the feature matrix and the adjacency matrix, where $x_i \in R^F$ is the feature vector of $v_i$ and $A_{ij} = 1$ if $(v_i, v_j) \in E$. $p_i = \{v_j \mid j \neq i\}$ represents the positives selected according to probability. We learn a GNN encoder $f(X, A) \in R^{N \times F'}, F' << F$ to embedding nodes representation without label information into a low-dimensional space, and then

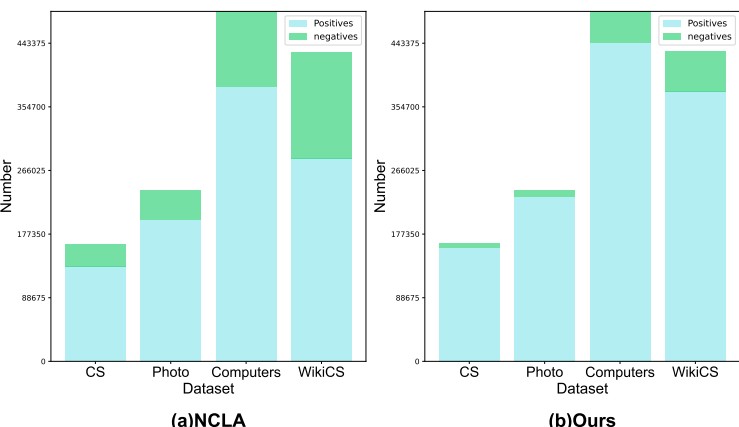

**(a)NCLA**    **(b)Ours**

Figure 2: Blue is positive, green is negative. (a) shows the number of neighbors that are actually positives or negatives with the anchor in the NCLA method and (b) show the same number of nodes selected in our method as the neighbors. The number of nodes that are positives or negatives with anchors.

apply the low-dimensional embedding representation to downstream tasks including node classification. And we sample two augmentation functions $t_1 \sim T$ and $t_2 \sim T$ from the set of all augmentation functions T. Then we get two augmentation views from G, $\tilde{G}_1 = t_1(G)$ and $\tilde{G}_2 = t_2(G)$. Given $\tilde{G}_1 = (\tilde{X}_1, \tilde{A}_1)$ and $\tilde{G}_2 = (\tilde{X}_2, \tilde{A}_2)$, we employs the GNN encoder to learn the embeddings $H^{(1)} = f(\tilde{X}_1, \tilde{A}_1) \in R^{N \times F'}$, $H^{(2)} = f(\tilde{X}_2, \tilde{A}_2) \in R^{N \times F'}$. For any node $\nu_i$, its embedding in one view $h_i^k$ is regarded as the anchor.

## 3.2 POSITIVE MINING

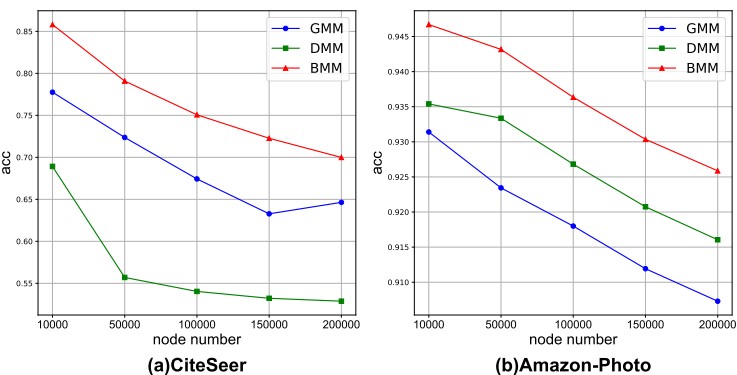

**(a)CiteSeer**    **(b)Amazon-Photo**

Figure 3: The proportion of true positive nodes over all sampled nodes when different numbers of positive nodes are sampled using different mixture models.

Graph Contrastive learning (GCL) effectively learns the representation of nodes by pulling pairs of positive nodes (or similar nodes) closer together in the representation space while separating pairs of negative nodes (or dissimilar nodes). However, many methods only pay attention to mining negative pairs, but ignore the role of positive pairs. We aim to select the positive nodes that are more reliably associated with anchors. In a recent study, NCLA (Shen et al., 2023) proposed that neighbor nodes be considered as positives for the anchor and the remaining nodes be the negatives of anchors. However, according to our experiments, as shown in Figure 2(a), there is still a large number of negatives in the neighbor nodes, which causes the negatives to be mistakenly pulled closer to the anchor. Actually, the negatives should be away from the anchor. In order to select more

correct positives, we believe that a mixture model (Lindsay, 1995) can be employed to estimate the probability of other nodes being positive with the anchor. We used three different Mixture models for experiments, Gaussian Mixture Model (GMM) (Reynolds et al., 2009) , Beta Mixture Model (BMM) (Gupta & Nadarajah, 2004; Ji et al., 2005; Antoniak, 1974) and Dirichlet Mixture Model (DMM) (Minka, 2000; Pitman & Yor, 1997). We also experiment that the Gaussian mixture model and Dirichlet mixture model have lower accuracy than BMM while obtaining the same number of positives. As shown in Figure 3 , an increase in the number of nodes correlates with a decline in the accuracy of correctly identifying positive nodes across all three mixture models. Notably, GMM shows higher accuracy than DMM on CiteSeer, while DMM outperforms GMM on Amazon-Photo dataset. However, the accuracy of Bayesian mixture model (BMM) is significantly better than that of Gaussian Mixture model (GMM) and Dirichlet mixture model (DMM) on CiteSeer and Amazon-Photo datasets. Therefore, we use the beta distribution, which can obtain more accurate positive nodes. Additionally, as shown in Figure 2 (b), by calculating the probability between positive samples and anchors by BMM, our method can obtain more accurate true positive samples of anchors. Also, we compare the performance of BMM, GMM and DMM in Table 3 and find that BMM consistently outperforms GMM and DMM. We adopt a C-component, C=2, BMM to model the distribution of true positives and false positives. The probability density function (pdf) of the beta distribution is:

$$p(s \mid \alpha, \beta) = \frac{\Gamma(\alpha + \beta)}{\Gamma(\alpha)\Gamma(\beta)} s^{\alpha-1}(1 - s)^{\beta-1} \tag{1}$$

The pdf of the s (Min-Max normalized cosine similarity of the two-component beta mixture model in the node normalized embeddings) can be defined as:

$$p(s) = \sum_{i=1}^{2} \lambda_i p(s \mid \alpha_i, \beta_i) \tag{2}$$

Where $\lambda_i$ is the mixture coefficients. Then we fit a two component BMM to model the distribution of true and false positives and we utilize Expectation Maximization (EM) algorithm to fit BMM.

In E-step, we fix the parameters of BMM ($\lambda_i, \alpha_i, \beta_i$) and update $p(c \mid s)$ with Bayes rule,

$$p(c \mid s) = \frac{\lambda_c p(s \mid \alpha_c, \beta_c)}{\sum_{i=1}^{C} \lambda_i p(s \mid \alpha_i, \beta_i)}, c = 1, \cdots, C \tag{3}$$

However, Fitting the Beta Mixture Model (BMM) with all similarity measures, especially on large datasets, can incur high computational costs. Therefore, we fit the BMM using random partial sampling and similarity assessments to reduce computational expenses. Then we get the weight average $\bar{s}_c$ and variance $\nu_c^2$ in sampling M similarities,

$$\bar{s}_c = \frac{\sum_{i=1}^{M} p(c \mid s_i) s_i}{\sum_{i=1}^{M} p(c \mid s_i)}, \nu_c^2 = \frac{\sum_{i=1}^{M} p(c \mid s_i)(s_i - \bar{s}_c)}{\sum_{i=1}^{M} p(c \mid s_i)} \tag{4}$$

In M-step, the parameter $\lambda_i$, $\alpha_i$ and $\beta_i$ of the model are estimated by the method of moments of statistics,

$$\alpha_i = \bar{s}_i \left( \frac{\bar{s}_i(1 - \bar{s}_i)}{\nu_i^2} - 1 \right) \tag{5}$$

$$\beta_i = \frac{\alpha_i (1 - \bar{s}_i)}{\bar{s}_i} \tag{6}$$

$$\lambda_i = \frac{1}{M} \sum_{j=1}^{M} p(i \mid s_j) \tag{7}$$

The training of the BMM occurs independently during one of the epochs in the model training process, rather than following the training of the entire model. Finally,we can decide whether two nodes are positive or not based on s (the similarity between them), the probability function is

$$p(c \mid s) = \frac{\lambda_c p(s \mid \alpha_c, \beta_c)}{p(s)} \tag{8}$$

By using the posterior probability calculated by the EM algorithm, more accurate positives of anchor points can be obtained, and then contrastive learning can be performed.

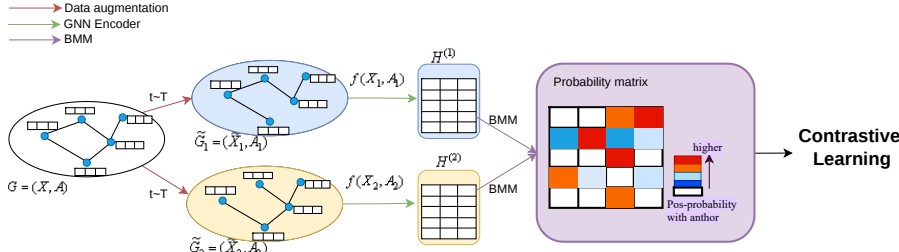

Figure 4: The model architecture of PMGCL. It generates two views and trains the BMM with the similarity of the two views after encoding by the GNN encoder to generate a probability matrix. The probability of its own node is removed from the probability matrix, since different view embedding representations of the same node must serve as positive pairs. Then the k nodes with the highest probability are sampled from the probability matrix as positives.

### 3.3 MULTIPLE POSITIVE CONTRASTIVE LEARNING

GCL maximizes Mutual Information (MI) by contrasting positive and negative pairs. InfoNCE and NT-Xent are widely used in node-node contrastive learning (Wan et al., 2021a;b; Xia et al., 2022). However, this approach only one positive pair per anchor exists, where node embeddings in different views are defined as positive pairs, will cause other positive pairs to move undesirably away from the anchor. To solve this problem, as shown in Figure 4, we use BMM to estimate the probability of other nodes being positive with the anchor, from this, the k nodes with the highest probability are obtained as the positive pairs for the anchor. Let $h_i^{(1)}$ and $h_i^{(2)}$ denote the embeddings of $v_i$ learned by view1 and view2 respectively. Selecting $h_i^{(1)}$ as the anchor, the positives come from three sources: 1) inter-view same node, such as the embedding of the same node in different view $h_i^{(2)}$. 2) intra-view nodes selected by BMM and 3) inter-view nodes selected by BMM. Therefore, the number of positive pairs of anchors is 2k+1, where k is the number of positive pairs obtained by BMM. The contrastive loss function of the anchor $h_i^{(2)}$ is expressed as:

$$\ell(h_i^{(1)}) = -\log \frac{(e^{\theta(h_i^{(1)}, h_i^{(2)})/\tau} + \sum_{j=1}^{k}(e^{\theta(h_i^{(1)}, h_j^{(1)})/\tau} + e^{\theta(h_i^{(1)}, h_j^{(2)})/\tau}))/(2k+1)}{(e^{\theta(h_i^{(1)}, h_i^{(2)})/\tau} + \sum_{j \neq i}(e^{\theta(h_i^{(1)}, h_j^{(1)})/\tau} + e^{\theta(h_i^{(1)}, h_j^{(2)})/\tau}))} \quad (9)$$

where $\tau$ is a temperature parameter, and $\theta(x, y)$ is the similarity between x and y. Decompose Eq(9) the last two terms:

$$\sum_{j \neq i} e^{\theta(h_i^{(1)}, h_j^{(1)})/\tau} = \underbrace{\sum_{\nu_i \in K} e^{\theta(h_i^{(1)}, h_j^{(1)})/\tau}}_{intra-view \quad pos} + \underbrace{\sum_{\nu_i \notin K} e^{\theta(h_i^{(1)}, h_j^{(1)})/\tau}}_{intra-view \quad neg} \quad (10)$$

$$\sum_{j \neq i} e^{\theta(h_i^{(1)}, h_j^{(2)})/\tau} = \underbrace{\sum_{\nu_i \in K} e^{\theta(h_i^{(1)}, h_j^{(2)})/\tau}}_{inter-view \quad pos} + \underbrace{\sum_{\nu_i \notin K} e^{\theta(h_i^{(1)}, h_j^{(2)})/\tau}}_{inter-view \quad neg} \quad (11)$$

Where K is the set of k positive nodes obtained from BMM. Minimizing Eq(9) will maximize the MI between positive pairs and minimize the MI between negative pairs. This loss function is an evolved version of the NT-Xent loss, where only one positive pair exists. Since the two views are symmetric, the loss function $\ell(h_i^2)$ can be similarly defined as Eq(9) for the embedded $h_i^{(2)}$ of view 2 of a given $\nu_i$ as an anchor. Finally the combination of view 1 and the view 2 loss is defined as:

$$L(H^{(1)}, H^{(2)}) = \frac{1}{2N} \sum_{i=1}^{N} [\ell(H^{(1)}) + \ell(H^{(2)})] \quad (12)$$

Since both inter-view and intra-view contain all negative pairs, we only need to fit the BMM with similarity from a single view. In our experiments, we only use the similarity of the interview perspective to fit the BMM. Algorithm 1 summarizes the training algorithm of PMGCL for the node classification task, please refer to appendix C for details.

### 3.4 Time Complexity Analysis

In the training process, using BMM to estimate the probability takes a slight time, but we only need to fit the BMM once in the whole training process, instead of fitting each epoch once, and we obtain M $\left(M << N^2\right)$ similarities to fit the BMM by random sampling method. Therefore, the time complexity of fitting BMM by EM algorithm is $O(IM)$. I is the number of iterations that fit the BMM. The time complexity of neighbor contrastive learning is $O(N^2 F')$ where N is the number of nodes. F is the number of input features and $F'$ is the embedding dimension. Thus, the total time complexity is $O(IM + N^2 F')$.

## 4 Experiments

### 4.1 Experiments Setup

**Datasets.** We evaluate our method using seven widely used datasets, Cora, Citeseer, and PubMed from the Plantoid (Kipf & Welling, 2016), Photo and Computers from the Amazon (McAuley et al., 2015), a co-authorship network Coauthor-CS (Shchur et al., 2018), a reference network from Wikipedia WikiCS (Mernyei & Cangea, 2007). More details are in the appendix A.

**Baselines.** We primarily compare our PMGCL with classical GSSL algorithms: DeepWalk (Perozzi et al., 2014) and Node2vec (Grover & Leskovec, 2016). Additionally, we also consider other recent GSSL baselines: BGRL (Thakoor et al., 2021), GRACE (Zhu et al., 2020), GCA (Zhu et al., 2021), MVGRL (Hassani & Khasahmadi, 2020), DGI (Veličković et al., 2018), GBT (Bielak et al., 2022), ProGCL (Xia et al., 2021b) and PiGCL (He et al., 2024). We also compare PMGCL with supervised counterparts including GCN (Kipf & Welling, 2016) and Graph Attention Networks (GAT) (Veličković et al., 2017).

**Detailed Setup.** Based on previous work (Veličković et al., 2018), we trained the model in an unsupervised manner. We test our PMGCL on classification and clustering tasks. For classification task, we adopt a two-layer GCN (Kipf & Welling, 2016) as transduction study of encoder. We follow the GRACE test (Zhu et al., 2020). Specifically, we use 10% of the data to train the downstream classifier and the remaining 90% for testing. We run it 20 times and then report the average accuracy. For the clustering task, we directly feed the obtained representation into a randomly initialized K-Means (Macqueen, 1967) predictor. We run 10 times and report the average NMI and ARI.

### 4.2 Performance Analysis

Table 1: Accuracy(± std) on the node classification task. The best and second best results are highlighted in boldface and underlined, respectively.

| Method | Cora | CiteSeer | PubMed | CS | Photo | Computers | WikiCS |
|--------|------|----------|--------|-----|-------|-----------|--------|
| Rf | 64.80 | 64.60 | 84.80 | 90.37 | 79.53 | 73.81 | 71.98 |
| N2v | 74.80 | 52.30 | 80.30 | 85.08 | 89.67 | 84.39 | 71.79 |
| DW | 75.70 | 50.50 | 80.50 | 84.61 | 89.44 | 85.68 | 74.35 |
| DW+F | 73.10 | 47.60 | 83.70 | 87.70 | 90.05 | 86.28 | 77.21 |
| BGRL | 81.30 ± 0.31 | 70.57 ± 0.98 | 85.86 ± 0.15 | 92.37 ± 0.22 | 92.36 ± 0.09 | 87.28 ± 0.34 | 78.41 ± 0.09 |
| GRACE | 83.30 ± 0.40 | 71.65 ± 1.03 | 85.69 ± 0.20 | 92.06 ± 0.18 | 92.13 ± 0.20 | 87.13 ± 0.37 | 77.97 ± 0.63 |
| GCA | 83.42 ± 0.78 | 70.79 ± 1.32 | 86.12 ± 0.22 | 93.01 ± 0.21 | 92.15 ± 0.26 | 88.04 ± 0.34 | 77.94 ± 0.67 |
| MVGRL | 84.32 ± 0.94 | 72.29 ± 0.75 | 85.33 ± 0.25 | 92.28 ± 0.19 | 92.07 ± 0.26 | 87.68 ± 0.31 | 77.52 ± 0.08 |
| DGI | 83.24 ± 0.73 | 71.91 ± 0.85 | 85.67 ± 0.28 | 92.86 ± 0.15 | 92.56 ± 0.41 | 86.93 ± 0.25 | 75.35 ± 0.14 |
| GBT | 83.50 ± 1.05 | 69.12 ± 1.39 | 85.29 ± 0.34 | 92.63 ± 0.14 | 92.52 ± 0.34 | 87.13 ± 0.37 | 76.65 ± 0.62 |
| ProGCL | 83.12 ± 0.78 | 72.85 ± 0.92 | 85.60 ± 0.15 | 93.24 ± 0.20 | 93.03 ± 0.13 | 87.65 ± 0.21 | 78.51 ± 0.12 |
| PiGCL | 84.62 ± 0.62 | 72.86 ± 0.46 | 86.68 ± 0.06 | 93.21 ± 0.09 | 93.01 ± 0.08 | 88.81 ± 0.27 | 78.34 ± 0.26 |
| PMGCL | **86.60 ± 0.13** | **73.82 ± 0.18** | **86.95 ± 0.05** | **93.33 ± 0.13** | **93.27 ± 0.11** | **89.51 ± 0.14** | **79.64 ± 0.04** |
| GCN | 82.80 | 72.00 | 84.80 | 93.03 | 92.42 | 86.51 | 77.19 |
| GAT | 83.00 | 72.50 | 79.00 | 92.31 | 92.56 | 86.93 | 77.65 |

Table 2: Performance on node clustering.

|          |     | BGRL   | DGI    | GRACE  | GBT    | GCA    | ProGCL | PiGCL  | PMGCL  |
|----------|-----|--------|--------|--------|--------|--------|--------|--------|--------|
| Cora     | NMI | 0.4211 | 0.5370 | 0.4758 | 0.4562 | 0.4510 | 0.5131 | 0.5275 | **0.5447** |
|          | ARI | 0.2905 | 0.4469 | 0.3633 | 0.3683 | 0.3104 | 0.3434 | 0.4604 | **0.4970** |
| CiteSeer | NMI | 0.3748 | 0.4185 | 0.3960 | 0.3414 | 0.3737 | 0.4115 | **0.4515** | 0.4501 |
|          | ARI | 0.3855 | 0.4140 | 0.3977 | 0.3193 | 0.3675 | 0.4219 | 0.4611 | **0.4679** |
| PubMed   | NMI | 0.3149 | 0.3188 | 0.3508 | 0.2992 | 0.3307 | 0.3595 | 0.3542 | **0.3625** |
|          | ARI | 0.2928 | 0.3165 | 0.3286 | 0.2942 | 0.2919 | 0.3264 | **0.4055** | 0.3319 |
| Photo    | NMI | 0.6189 | 0.3764 | 0.5346 | 0.5847 | 0.6147 | 0.6122 | 0.5409 | **0.6443** |
|          | ARI | 0.4754 | 0.2643 | 0.4247 | 0.4702 | 0.4943 | 0.4653 | 0.4524 | **0.5262** |

**Classification.** For the classification task, as shown in Table 1, on the seven data sets, our proposed PMGCL consistently performs optimal results for both unsupervised and supervised baselines, which verifies the superiority of our PMGCL. Our observations are as follows: First, traditional methods node2vec(for short "N2v") and DeepWalk(for short "DW") which solely rely on adjacency matrices, outperform basic logistic regression classifiers that utilize raw features ("raw features" for short "Rf") across the Cora, Citeseer and Amazon datasets. However, the latter performs better on the other three datasets. Combing the both ("DeepWalk + features" for short "DW+F") can bring significant improvements. Compared with the model using a single positive pair, our PMGCL obtains more reasonable positive nodes as the positive nodes of the anchor. This enables the propagation of scarce label information by utilizing appropriate positive and negative samples in the absence of labels, thereby effectively enhancing node classification performance.

**Clustering.** In the clustering task, we evaluated PMGCL on the Cora, CiteSeer, Amazon-Photo and PubMed datasets, and we clustered the learned embeddings using the K-means algorithm. As shown in Table 2, our PMGCL again exhibits excellent performance. Compared to the case where there is only one positive pair, we increase the number of positive pairs by obtaining suitable positive pairs, which pulls the positives closer to the anchor and helps the clustering task. It shows that PMGCL's strategy of obtaining positive pairs is successful.

## 4.3 PARAMETERS ANALYSIS

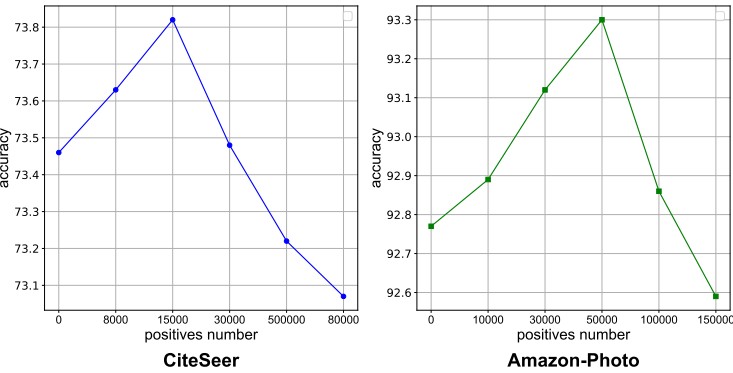

Figure 5: Sensitivity analysis of the hyperparameter k on PMGCL.

We experiment the model performance on CiteSeer and Photo datasets under different hyperparameter settings for the number of positive nodes, denoted as k. As shown in Figure 5, when k is within a certain range, the accuracy is greater than that when k=0. This indicates that appropriately increasing the number of positives can improve the performance of the model, and the model overall exhibits a unimodal pattern. With the increase of k, the model obtains more positives. However, when k is too large, as shown in Figure 3, the proportion of true positives decreases, the benefit brought by the positives becomes limited, and the number of false positives increases, leading to a decrease in model performance, even lower than when k=0. Therefore, to maximize the model performance, the

number of obtained positives should be at a suitable intermediate value. In the appendix B we show more parametric analysis.

### 4.4 ABLATION STUDY

In this section, we replace or remove various parts of PMGCL, studying the impact of each component.

Table 3: Comparison of BMM, GMM, and DMM.

|  | Photo | Computers | Coauthor-CS |
| --- | --- | --- | --- |
| GMM | 92.94 | 88.85 | 93.02 |
| BMM | 92.33 | 89.47 | 92.88 |
| DMM | **93.37** | **89.54** | **93.33** |

As shown in Table 3, we replaced BMM with GMM and DMM, respectively, and compared the performance on Photo, Computers and Coauthor-CS datasets. Combined with Figure 5, is able to select more accurate positive samples and demonstrates better performance.

Table 4: Comparison of different contrastive loss functions.

|  | Photo | Computers | Coauthor-CS |
| --- | --- | --- | --- |
| InfoNCE | 93.05 | 89.21 | 93.02 |
| NT-Xent | 92.88 | 89.17 | 93.05 |
| PMGCL | **93.37** | **89.54** | **93.33** |

Next, we replaced the contrastive loss of multi-positive nodes with the single positive pair pattern used in InfoNCE and NT-Xent. As shown in Table 4, the proposed neighbor contrastive loss consistently achieves the highest accuracy across all loss variants for the three datasets. This indicates the effectiveness of the positive nodes we selected.

## 5 CONCLUSIONS

To address the challenges of difficult positive selection in GCL and the insufficiency of positives in contrastive loss, PMGCL has been proposed. On one hand, PMGCL can more accurately select positive nodes by fitting the BMM. On the other hand, instead of directly adopting the contrastive loss from computer vision, PMGCL improves upon it by transitioning from a single positive to multiple positives. Consequently, PMGCL significantly enhances the utilization of positive nodes. Extensive experiments in node classification and clustering demonstrate that the PMGCL method can identify a greater number of more accurate positive nodes and can achieve superior performance across multiple tasks.

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

## A  DATASETS

Table 5: Statistics of datasets used in experiments.

| Datasets | Nodes | Edges | Features | Labels |
|---|---|---|---|---|
| Cora | 2708 | 10556 | 1433 | 7 |
| CiteSeer | 3327 | 9228 | 3703 | 6 |
| PubMed | 19717 | 88651 | 500 | 3 |
| Amazon-Photo | 7650 | 238162 | 745 | 8 |
| Amazon-Computers | 13752 | 245861 | 767 | 10 |
| Coauthor-CS | 18333 | 163788 | 6805 | 15 |
| Wiki-CS | 11701 | 216123 | 300 | 10 |

We introduce the dataset used for our experiments as follows:

- **Cora** (Kipf & Welling, 2016) is a scientific literature network dataset where nodes represent scientific papers and edges represent citation relationships between papers.Each node has a set of features, usually a bag-of-words representation, and a category label.

- **CiteSeer** (Kipf & Welling, 2016) Like Cora, the CiteSeer dataset also contains a network of scientific literature, where the nodes are papers and the edges are citation relationships. The feature of a paper can be a vector representation of keywords, abstract or full text.

- **PubMed** (Kipf & Welling, 2016) is a large citation network dataset of biomedical literature. It contains paper nodes and citation edges, as well as the title, abstract, and keywords of the paper.

- **Amazon-computer and Amazon-Photo** (McAuley et al., 2015) are from the Amazon product co-occurrence network, where nodes represent products and edges represent co-occurrence relationships between products. Each node has a sparse bag-of-words feature that encodes a product review and is labeled with its category.

- **Coauthor-CS** (Shchur et al., 2018) is based on collaborations between researchers in the field of computer science, where nodes are researchers and edges represent collaborations between two researchers. Each node has a bag-of-words feature based on the keywords of the author's paper. The tagging of authors is their most active research area.

- **WikiCS** (Mernyei & Cangea, 2007) contains a network of computer science-related pages on Wikipedia, where nodes are pages and edges are links between pages. The nodes are divided into ten classes, each representing a branch of the field.

## B   HYPER-PARAMETERS ANALYSIS

In Figure 6, we further investigate the impact of the number of iterations I and the number of samples M of the EM algorithm. As shown in Figure 6(a), when the number of iterations of our EM algorithm increases, we can observe that the accuracy is flat or there is a small improvement. However, this would introduce more computational overhead; therefore, we set I = 10 in our experiments. As shown in Figure 6(b), more similarities are sampled by fitting BMM. With the increase of the number of samples, the accuracy is maintained within a certain range, and the improvement effect is not obvious. However, it incurs more computational overhead. Therefore, in our experiments, we only sample = 100 samples for each anchor.

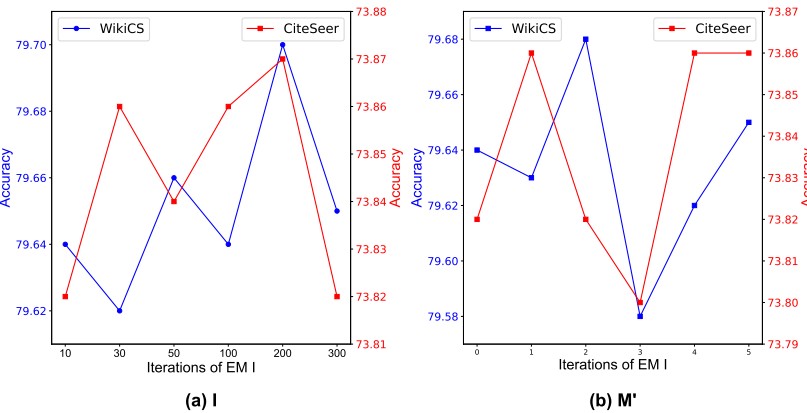

Figure 6:  Accuracy when varying I and $M'(M = NM')$.

## C   PSEUDO CODES OF PMGCL

---
**Algorithm 1** PMGCL
---
**Input:** T,G,f,N,normalized cosine similarity s, epoch for fitting BMM E, selective positive number k.

 1: **for** epoch = 0,1,2... **do**
 2:     Generate two augmented functions $t_1 \sim T$, $t_2 \sim T$
 3:     $\tilde{G}_1 = t_1(G), \tilde{G}_2 = t_2(G)$;
 4:     $H^{(1)} = f(\tilde{X}_1, \tilde{A}_1), H^{(2)} = f(\tilde{X}_2, \tilde{A}_2)$;
 5:     **for** $h_i^{(1)} \in H^{(1)}$ and $h_i^{(2)} \in H^{(2)}$ **do**
 6:         $s_{ij} = s(h_i^{(1)}, h_i^{(2)})$
 7:        **if** epoch=E **then**
 8:            Compute $p(c \mid s_{ij})$ with Eq(1) to Eq(8).
 9:        **end if**
10:     **end for**
11:     **if** $Epoch \geq E$ **then**
12:         Select k positive subsets.
13:         Compute contrastive loss L with Eq(12)
14:         Update the parameters of $f$ with L
15:     **else**
16:         Compute contrastive loss L with InfoNCE
17:         Update the parameters of $f$ with L
18:     **end if**
19: **end for**
20:
**Output:** f

---

