# OpenReview forum: "Positive Mining in Graph Contrastive Learning"
_ICLR.cc/2025/Conference — ICLR 2025 Conference Withdrawn Submission_

### Official Review · Reviewer_XoSq · 2024-10-30

**Soundness:** 3
**Presentation:** 3
**Contribution:** 2
**Rating:** 6
**Confidence:** 5

**Summary:**

Summary:
Graph Contrastive Learning (GCL) has achieved notable advancements. Recent research has focused on refining these loss functions by adjusting the weighting of negative nodes or using similarity measures to select positive nodes for anchor nodes. However, there is a concern that the current positive and negative nodes may not accurately reflect true relational patterns. To address this, the author proposed a new approach called PMGCL , which utilizes a mixture model to calculate the probability of positive samples, helping to identify true positive nodes more accurately in relation to the anchor node. Evaluations on real-world graph datasets show that PMGCL surpasses traditional GCL methods.

**Strengths:**

Pro:
- The paper proposed a new approach called PMGCL , which utilizes a mixture model to calculate the probability of positive samples.
- The paper achieved SOTA performance compared with baselines

**Weaknesses:**

Cons:
- The proposed contrastive learning methods seem like an advancement based on node level Info-NCE losses. However, contrastive learning methods still include other kinds of losses like DGI[1] loss, barlow twins losses[2] and graph group discrimination[2] loss. Can this positive mining method be adapted and benefit from these losses?
- Scalability of the model is encouraged to be evaluated. Ogbn-arxiv level dataset evaluation is necessary.

**Questions:**

same as cons

---

### Official Review · Reviewer_1Ca4 · 2024-10-31

**Soundness:** 2
**Presentation:** 1
**Contribution:** 1
**Rating:** 3
**Confidence:** 5

**Summary:**

This article focuses on self-supervised learning at the node level in graph-structured data, particularly contrastive learning. In traditional graph contrastive learning, false positives often occur, leading to suboptimal performance. This paper uses a beta-mixture model to calculate the probability of an anchor node and other nodes being positive examples, thereby filtering out true positives from all positive candidate pairs.

**Strengths:**

1.	I appreciate the research problem addressed in this paper. Better distinguishing between positive and negative examples in graph contrastive learning is indeed very important in graph contrastive learning research.

**Weaknesses:**

1.	The paper is poorly written overall. For example, the introduction of the proposed method is not very clear. For instance, 's' in equation (1) should have a more specific mathematical expression. Mathematical notations are not standardized, such as matrix expression $ \mathbf{X}$ and real number space expression $\mathbb{R}$. The quality of figures, such as Figure 2 and Figure 3, needs improvement.
2.	The paper lacks novelty. The main contribution is using BMM for hard (true) positive mining. However, to my knowledge, this method has already been used in Graph Contrastive Learning in ProGCL.
3.	The experimental results are not sufficient and convincing. The paper claims to achieve SOTA performance on node classification tasks. However, the baseline results seem unreliable to me (I think they were underreported) - they appear to be results obtained on public splits rather than the 10% training set split.
4.	In my view, the computational complexity of this method should be very high, yet the authors failed to provide actual training time comparisons.

**Questions:**

1.	In Section 3.4, the authors mention they "only need to fit the BMM once." I don't quite understand this because according to equation 1, s is the cosine similarity of node embeddings. Since embeddings change during training, I think it would make more sense for the BMM to change accordingly.
2.	What exactly does the "node number" mentioned in many places (such as in Figure 3) refer to? According to my understanding, it actually refers to the number of positive pairs?
3.	What is the performance on public splits for citation graphs?
4.	What is the actual running time of PMGCL compared to other methods, especially on large graphs?

---

### Official Review · Reviewer_fmcg · 2024-11-02

**Soundness:** 2
**Presentation:** 1
**Contribution:** 2
**Rating:** 3
**Confidence:** 4

**Summary:**

In this paper, the authors introduce Positive Mining Graph Contrastive Learning (PMGCL), which employs a mixture model to estimate the likelihood of positive samples relative to an anchor node. Their experiments demonstrate that PMGCL significantly outperforms traditional GCL methods, achieving state-of-the-art results in unsupervised benchmarks and surpassing some supervised learning performances.

**Strengths:**

1. Enhanced positive pair selection: By employing a Bayesian Mixture Model (BMM) to estimate the probability of other nodes being true positives relative to the anchor node, the proposed method provides a more effective way to select positive pairs.

2. Improved contrastive loss framework: Instead of applying traditional contrastive loss to graph data, the authors introduce a new loss function that accommodates multiple positive pairs per anchor, leading to significant improvements over existing GCL methods. The proposed approach achieves superior performance on node classification tasks across various unsupervised datasets and even exceeds some supervised benchmarks, while also showing promising results in node clustering tasks.

**Weaknesses:**

1. The experiments lack comparisons with state-of-the-art (SOTA) methods. For example, the paper should include comparisons with GCL methods that utilize different data augmentation strategies, such as AutoGCL, which is based on learnable graph data augmentation, and GA2C.

2. Table 4 lacks analytical experiments on different graph contrastive losses. For instance, it would be beneficial to include losses like the graph Barlow Twins loss and point-to-point losses (such as BGRL loss).

3. The formalization is not rigorous. For example, the notation "k nodes" should have "k" in italics, "Eq" should be abbreviated as "Eq.", and "x and y" in "between x and y" should have "x" and "y" in italics (Line 304).

4. The quality of the paper needs improvement. For example, the presentation of Figure 4 is poor. In Line 290, the phrase "this approach only one positive pair per anchor exists" should be revised to "there is only one positive pair per anchor in this approach." Additionally, in Line 304, "Decompose Eq(9) the last two terms" should be revised to "Decompose the last two terms in Eq. (9)."

**Questions:**

Please see my previous comment.

---

### Official Review · Reviewer_D27i · 2024-11-03

**Soundness:** 3
**Presentation:** 4
**Contribution:** 3
**Rating:** 5
**Confidence:** 4

**Summary:**

This paper proposes a new method, Positive Sample Mining Graph Contrastive Learning (PMGCL), to address the problem of inaccurate selection of positive and negative samples in Graph Contrastive Learning (GCL). This method uses a Bayesian mixture model (BMM) to calculate the positive sample probability between the anchor node and other nodes, thereby more accurately identifying the positive samples that are truly related to the anchor node. Experiments show that PMGCL outperforms existing GCL methods in node classification tasks on multiple unsupervised datasets.

**Strengths:**

1.The paper proposes a new graph contrastive learning method PMGCL, which improves the accuracy of learning by mining positive samples, providing a new idea for graph contrastive learning.

2.The paper clearly explains the principle and implementation steps of the PMGCL method, making it easier for readers to understand and reproduce the method.

3.The paper conducts a detailed analysis of the computational complexity of PMGCL, showing that it does not bring excessive computational burden, which is of great significance for practical applications.

**Weaknesses:**

1.The paper's survey of related work appears somewhat outdated. While the proposed PMGCL method is indeed innovative in its approach to enhancing the accuracy of graph contrastive learning through positive sample mining, the cited references and comparisons to existing methods do not adequately cover the most recent advancements in this field. Incorporating more recent research would provide a more comprehensive context for evaluating the novelty and significance of PMGCL.

2.The experiment lacks comparison with the most recent comparative baselines (e.g., ProGCL[1], AFGRL [2], iGCL [3]). Although the PMGCL method proposed in the paper demonstrates some performance improvement on node classification and clustering tasks, however, the failure to conduct a comprehensive and in-depth comparison with the current state-of-the-art makes it difficult to assess the superiority of the PMGCL method in a rigorous and comprehensive manner.

3.The paper lacks adequate theoretical analysis, particularly in explaining the advantages of PMGCL over other approaches, such as those that focus on expanding negative samples. A more in-depth discussion on how PMGCL's strategy of utilizing a mixture model to identify true positives improves over existing methods would be beneficial. Additionally, the paper does not provide sufficient theoretical analysis of the proposed contrastive loss function that accommodates multiple positive pairs per anchor. A clearer explanation of how this loss function differs from and improves upon traditional contrastive loss functions would greatly enhance the paper's theoretical contributions.


[1] Jun Xia, Lirong Wu, Ge Wang, Jintao Chen, Stan Z. Li: ProGCL: Rethinking Hard Negative Mining in Graph Contrastive Learning. ICML 2022
[2]Namkyeong Lee, Junseok Lee, Chanyoung Park: Augmentation-Free Self-Supervised Learning on Graphs. AAAI 2022
[3]Haifeng Li, Jun Cao, Jiawei Zhu, Qinyao Luo, Silu He, Xuying Wang: Augmentation-Free Graph Contrastive Learning of Invariant-Discriminative Representations. IEEE TNNLS 2024

**Questions:**

1.The proposed method designs multiple positive samples. Could you further analyze the advantages and essential differences compared to the method of expanding negative samples?

2.Is the proposed loss function applicable to other graph contrastive learning methods?

3.How is the ratio of positive and negative samples determined in PMGCL? Will this ratio have an impact on the results?

---

### Official Review · Reviewer_ziyB · 2024-11-04

**Soundness:** 2
**Presentation:** 2
**Contribution:** 2
**Rating:** 3
**Confidence:** 4

**Summary:**

This paper introduces Positive Mining Graph Contrastive Learning (PMGCL), a method that aims to enhance Graph Contrastive Learning (GCL) by refining positive sample selection. PMGCL utilizes a Beta Mixture Model (BMM) to estimate the probability of true positive nodes, aiming to improve the accuracy of positive sampling and achieve multiple positive pairs for each anchor node. The paper reports improved performance over several existing GCL methods across multiple benchmark datasets.

**Strengths:**

1. **Relevance**:

   Positive mining in GCL is an important problem, and the method’s application could benefit GCL tasks if optimized further.

2. **Diverse Datasets**:

   This paper conducts evaluations on a variety of graph datasets, including citation networks and co-authorship networks. This diverse testing demonstrates the general applicability of PMGCL across different types of graph data, lending credibility to the method’s robustness.

**Weaknesses:**

1. **Limited Novelty**:

   The Gaussian Mixture Model (GMM) , Beta Mixture Model (BMM) and Dirichlet Mixture Model (DMM) models used for positive sampling are well-established models, and applying them in GCL does not represent a significant theoretical contribution. The novelty of the approach is therefore limited.

2. **Presentation Issues**:

   The paper’s presentation contains several spelling and grammatical errors, which impact readability. Some explanations are unclear, making it difficult to follow certain technical details.

   For example, in Figure 4, "Pos-probability with anthor" should be corrected to "anchor"; the image reference in line 227 should point to the subfigure in Figure 2(b) rather than Figure 2 as a whole; and in Algorithm 1, the formatting for "epoch comparison" should be consistent between lines 7 and 11.

3. **Outdated Baselines**:

   The experimental section primarily compares PMGCL with older baselines, with only one (PiGCL [1]) from the last two years. This choice weakens the paper’s claims of state-of-the-art performance, as more recent methods may perform differently on the same tasks.

4. **Lack of Theoretical Analysis**:

   The authors present a relatively "heuristic" approach to positive sampling, lacking robust theoretical support.

   Although the authors claim that the method has low complexity, they neither theoretically compare its overall complexity with other baselines nor experimentally validate its efficiency.



[1] Dongxiao He, Jitao Zhao, Cuiying Huo, Yongqi Huang, Yuxiao Huang,  and Zhiyong Feng. A new mechanism for eliminating implicit conflict in  graph contrastive learning. In Proceedings of the AAAI Conference on  Artificial Intelligence, volume 38, pp. 12340–12348, 2024.

**Questions:**

1. We would appreciate if the authors could provide a comparison of model training and inference times before and after applying this positive sampling strategy.
2. As ICLR is a high-level conference, we encourage the authors to include more theoretical proofs to substantiate the contributions of this work. Are there specific technical reasons for choosing BMM over other probabilistic models for positive sampling?

3. We hope the authors can include additional recent baselines in graph contrastive learning from the past two years, including but not limited to ImGCL [1], MA-GCL [2], GraphACL [3], and ASP [4].



[1] Zeng, Liang, et al. "Imgcl: Revisiting graph contrastive learning on imbalanced node classification." *Proceedings of the AAAI Conference on Artificial Intelligence*. Vol. 37. No. 9. 2023.

[2] Gong, Xumeng, Cheng Yang, and Chuan Shi. "Ma-gcl: Model augmentation tricks for graph contrastive learning." *Proceedings of the AAAI Conference on Artificial Intelligence*. Vol. 37. No. 4. 2023.

[3] Xiao, Teng, et al. "Simple and asymmetric graph contrastive learning without augmentations." *Advances in Neural Information Processing Systems* 36 (2024).

[4] Chen, Jialu, and Gang Kou. "Attribute and structure preserving graph contrastive learning." *Proceedings of the AAAI conference on artificial intelligence*. Vol. 37. No. 6. 2023.

---

### Note · Authors · 2024-11-26

I have read and agree with the venue's withdrawal policy on behalf of myself and my co-authors.